# Development of a Potency Assay for Nous-209, a Multivalent Neoantigens-Based Genetic Cancer Vaccine

**DOI:** 10.3390/vaccines12030325

**Published:** 2024-03-19

**Authors:** Rosa Bartolomeo, Fulvia Troise, Simona Allocca, Giulia Sdruscia, Rosa Vitale, Veronica Bignone, Anna Maria Petrone, Giuseppina Romano, Anna Morena D’Alise, Valentino Ruzza, Irene Garzia, Guido Leoni, Rossella Merone, Francesca Lanzaro, Stefano Colloca, Loredana Siani, Elisa Scarselli, Gabriella Cotugno

**Affiliations:** 1Nouscom Srl, Via di Castel Romano 100, 00128 Rome, Italy; r.bartolomeo@nouscom.com (R.B.); f.troise@nouscom.com (F.T.); s.allocca@nouscom.com (S.A.); g.sdruscia@nouscom.com (G.S.); r.vitale@nouscom.com (R.V.); v.bignone@nouscom.com (V.B.); a.petrone@nouscom.com (A.M.P.); g.romano@nouscom.com (G.R.); m.dalise@nouscom.com (A.M.D.); v.ruzza@nouscom.com (V.R.); i.garzia@nouscom.com (I.G.); g.leoni@nouscom.com (G.L.); l.siani@nouscom.com (L.S.); e.scarselli@nouscom.com (E.S.); 2ReiThera Srl, Via di Castel Romano 100, 00128 Rome, Italy; rossella.merone@reithera.com (R.M.); francesca.lanzaro@reithera.com (F.L.); stefano.colloca@reithera.com (S.C.)

**Keywords:** viral vectors, quality control testing, potency assessment, cancer vaccine, RT-Q-PCR potency assay, synthetic polypeptides

## Abstract

Quality control testing of vaccines, including potency assessment, is critical to ensure equivalence of clinical lots. We developed a potency assay to support the clinical advancement of Nous-209, a cancer vaccine based on heterologous prime/boost administration of two multivalent viral vector products: GAd-209 and MVA-209. These consist of a mix of four Adeno (Great Ape Adenovirus; GAd) and four Modified Vaccinia Ankara (MVA) vectors respectively, each containing a different transgene encoding a synthetic polypeptide composed of antigenic peptide fragments joined one after the other. The potency assay employs quantitative Reverse Transcription PCR (RT-Q-PCR) to quantitatively measure the transcripts from the four transgenes encoded by each product in in vitro infected cells, enabling simultaneous detection. Results showcase the assay’s robustness and biological relevance, as it effectively detects potency loss in one component of the mixture comparably to in vivo immunogenicity testing. This report details the assay’s setup and validation, offering valuable insights for the clinical development of similar genetic vaccines, particularly those encoding synthetic polypeptides.

## 1. Introduction

The measurement of the biological activity of vaccines is required for characterization of vaccine lots entering into the clinic. In vivo methods for vaccine potency testing require the sacrifice of large numbers of animals and suffer from long turn-around times and lack of reproducibility and precision caused by wide inter-animal variability. Thus, the use of in vitro assays as surrogate for the biological activity—immunogenicity—is preferred by both regulatory agencies and manufacturers. However, the development and validation of appropriate in vitro potency assays can be challenging, especially for multivalent genetic vaccines where the assay must be indicative of the potency of each vaccine component [1]. Given the complex nature of such products, results from multiple assays measuring different steps of the product biological activity are often preferred and defined as a potency assay matrix [2]. Specifically, for vaccines based on viral vectors, measurement of infectivity is often also part of the potency testing panel.

Relevant in vitro potency assays for genetic vaccines are expected to include quantitative measure of transgene expression [1]. Depending on factors related both to the nature of the tested vaccine and to possible technical limitations, different approaches have been applied and accepted for potency testing of approved vaccine products. As an example of vectored vaccines, potency testing for the Ad26.ZEBOV-GP Ebola vaccine (EMA/323670/2020) includes transgene expression analysis by Western blot, measurement of infectious units and ratio of virus particles/infectious units [3,4]. Similarly, potency of the MVA-BN-Filo Ebola vaccine lots (EMA/323668/2020) is addressed by measurement of the infectious titer and transgene expression on infected cells; this is accomplished by flow cytometry-based quantification of cells stained with antibodies against the virus or the transgene, respectively (EMA/323668/2020) [3,5]; Ad26.COV2.S COVID-19 vaccine potency testing includes transgene expression analysis by qualitative ELISA, measurement of infectious units and the ratio virus particles/infectious units [6]. Notably, development of a semi-quantitative or qualitative transgene expression test has been requested by regulatory agencies in addition to the proposed infectivity potency test to support batch release of the ChAdOx1-S COVID-19 vaccine (EMA/94907/2021 [7]).

Regulatory agencies require potency assays for release of ATMPs (Advanced Therapy Medicinal Products) in advanced phases of product development, to be correlated to the product mechanism of action. Thus, in the case of surrogate in vitro assays being used for product release, demonstration of the correlation of the selected assay with the biological activity is needed [8,9,10]. Notably, challenges for some gene therapy products under development have been reported [11,12].

Development of quantitative potency assays based on transgene expression detection can present difficulties for genetic vaccines encoding synthetic proteins for which antibodies are not available. This aspect is particularly relevant when the transgene encodes peptide fragments derived from different mutated proteins joined together [13,14,15,16,17,18]. This is the case for cancer vaccines encoding neoantigens, protein fragments including tumor-specific mutations, for which there is nowadays a great interest [19].

Specific and quantitative detection of transgene transcripts in infected cells can represent an alternative to the detection of the protein for those vaccines presenting limits in transgenes protein detection, provided that a link between transcript detection, protein expression and immune response induction is demonstrated. A similar strategy has been previously proposed for a plasmid-based vaccine product in clinical development (VCL-CB01 [20]).

Here we report our experience in the development of an in vitro potency assay based on quantitative transgene transcript detection for the characterization and release testing of Nous-209, an off-the-shelf cancer polyepitope vaccine based on viral vectors. Nous-209 is now being investigated in a phase 1/2 trial in patients with metastatic gastric, colorectal, and GEJ (gastroesophageal junction) mismatch repair deficient (dMMR) tumors combined with αPD-1 pembrolizumab (NCT04041310), and as a cancer-preventive vaccine in Lynch syndrome patients (NCT05078866).

## 2. Materials and Methods

### 2.1. Vaccine Vectors Production and Quantization

GAd-209-FSP and MVA-209-FSP vector composition has been already described in [21]. All the experiments described in the manuscript were conducted with purified GAd-209-FSP and MVA-209-FSP clinical lots, manufactured and released by Reithera s.r.l. (Rome, Italy) as previously described [13]. The impact of contaminants on the RT-Q-PCR testing readout was not assessed. The release test panel of clinical Drug Substances (DS) and Drug Products (DP) includes assessment of samples’ purity by testing residual host cell proteins (HEK-293 HCP ELISA kit, Cygnus Technologies, Inc., Southport, NC, USA) and host cell DNA (via specific qPCR assay). FSP stands for FrameShift Peptides. For generation of GAd-209-FSP new DPs in an R&D setting, the four clinical DSs (provided by Reithera s.r.l.) were diluted to a final concentration of 2 × 10^11^ vp/mL in formulation buffer (10 mM Tris, 10 mM Histidine, 8% Sucrose (*w*/*v*), 25 mM Sodium Chloride, 1 mM Magnesium Chloride, 0.02% Polysorbate 80 (PS-80) (*w*/*v*), pH 7.4) and mixed in a 1:1:1:1 ratio targeting a final concentration of 5 × 10^10^ vp/mL of each DS in the final DP mix (target titer of the DP is 2 × 10^11^ vp/mL). The titer was confirmed by transgene specific quantitative PCR analysis, in the absence of DNAse, of each DS in the DPs. Briefly, each DP was diluted 1:10 in 0.1% SDS in PBS and incubated for 10′ at 56 °C in a dry bath (Thermomixer compact, Eppendorf) to free the viral DNA. Each test sample was then further diluted 1:100 in water, molecular biology grade (GIBCO, Invitrogen Corporation, Carlsbad, CA, USA). An amount of 5 μL of the dilution was quantified by four transgene-specific qPCR reactions using the TaqMan Universal 2X PCR Master Mix (Applied Biosystems, ThermoFisher, Waltham, MA, USA) in a QuantStudio 12K Flex Real Time PCR System (Applied Biosystems). The sequence of transgene-specific oligos/probes used for each specific qPCR is reported in Appendix A. Absolute quantification of each DS component was performed using plasmid DNA standard curves for each FSP transgene.

### 2.2. Vaccine Vectors Thermal Inactivation

To assay potency loss, aliquots of GAd-209-FSP-A3 clinical DS, diluted to 2 × 10^11^ vp/mL, were incubated at 51 °C for 3 min or at 56 °C for 10 min before being mixed with the other 3 DS. Similarly, aliquots of MVA-209-FSP or GAd-209-FSP DPs (provided by Reithera s.r.l.) were incubated at 37 °C for 7 days or at 56 °C for 10 min. Incubations were performed in a dry bath (Thermomixer compact, Eppendorf) in a 1.5 mL plastic tube vial. For both infectivity and potency measurements, the potency loss was quantified by LRV (Log10 Reduction Value), corresponding to Log10 (Vi/Vni) where V indicates the value of either infectivity or potency, “i” indicates the inactivated sample and “ni” the non-inactivated sample. To allow LRV calculations for thermally inactivated samples whose FcDc (FSP cDNA copies) were undetectable, such values were approximated to 1 FcDc. For potency testing of new DPs produced in R&D settings, the GAd-209-FSP clinical DP was used as assay internal control, and results for each FSP-A transgene were expressed as percent of results obtained for the same transgene with the clinical lot.

### 2.3. Cell Culture

Hek-293, HeLa and Vero cell lines were purchased from ATCC; cells were grown in Dulbecco’s modified Eagle’s media (DMEM) supplemented with 10% (Hek293 and HeLa) and 5% (Vero) fetal bovine serum (FBS), 2 mM L-glutamine, and 100 U/mL penicillin/streptomycin (all reagents were purchased from GIBCO, Invitrogen Corporation, Carlsbad, CA, USA), at 37 °C in a 5% CO_2_ atmosphere. Cells were cultured in T-75 flasks and harvested for routine passage with a solution of 0.05% Trypsin/EDTA (GIBCO, Invitrogen Corporation, Carlsbad, CA, USA). Cells were split twice a week (at 1:3 dilution for Hek293; 1:6 dilution for Vero; 1:10 dilution for HeLa). All cell lines were used between passage 2 and passage 12 after thawing.

### 2.4. HeLa Cells Transduction

HeLa cells were seeded into six-well plates at a density of 1 × 10^6^ cells/well in 2 mL culture medium and allowed to adhere for 24 h at +37 °C. After 24 h, one well was detached and counted for exact MOI (multiplicity of infection) calculation. Cells were transduced in a total volume of 2 mL of fresh complete medium with either GAd-209-FSP (MOI: 20 to 160 vp/cell) or MVA-209-FSP (MOI: 0.1 to 0.9 ifu/cell) vectors (2 wells for each test item) and incubated at +37 °C. Cells were collected at the indicated time points for RNA extraction and transcript levels analysis. Before MVA transduction cells with media were maintained for 30′ in ice (synchronization) and right after addition of the viral vector dilutions, the cell plates were centrifuged for 30′ at 650 rcf at 4 °C (spinoculation) in a Beckman Coulter Allegra X-12R Refrigerated Centrifuge with Multiwell-plate carriers before incubation at +37 °C for the indicated time.

### 2.5. RNA Extraction and RT-Q-PCR

Cells were washed twice with Phosphate Buffer Saline (PBS, Gibco, Invitrogen Corporation, Carlsbad, CA, USA) and harvested using trypsin/EDTA (Gibco, Invitrogen Corporation, Carlsbad, CA, USA) for 3–5 min; then the cell suspension was centrifuged at 1200× *g* for 5 min, and the cell pellet washed with PBS and separated by centrifugation. The cell pellet was used to isolate total RNA using the QIAshredder and the RNeasy Mini kit (Qiagen, Hilden, Germany). Contaminating DNA was removed from each RNA extract by means of two DNase treatments. A first, on-column DNAse digestion with RNAse-Free DNase Set (Qiagen, Hilden, Germany) was performed during the RNA extraction by following the procedure suggested by the RNeasy Mini kit protocol. RNA was quantified with a Nanodrop 2000 microvolume spectrophotometer (Thermo Fisher Scientific, Waltham, MA, USA) and 1μg of total RNA was treated with 1 μL of DNase I, Amplification Grade (Invitrogen, ThermoFisher, Waltham, MA, USA) in a final volume of 10 μL in DNAse buffer (Invitrogen). The cDNA was synthesized using the SuperScript IV VILO Master Mix (Invitrogen, Thermo Fisher Scientific, Waltham, MA, USA), following the manufacturer’s recommendations, using 1 µg of DNAse-treated RNA as template. The same reaction but without RT enzyme was used to exclude non-specific amplification from residual viral DNA that may have resulted in “false-positive” signals. Such control was conducted on a subset of analyzed samples in each experiment to confirm absence of amplification signal. An amount of 5 μL of a 10X (GAd) or 5X (MVA) diluted cDNA was used as template for duplex qPCR by using the TaqMan Universal 2X PCR Master Mix (Applied Biosystems, ThermoFisher, Waltham, MA, USA) in a QuantStudio 12K Flex Real Time PCR System (Applied Biosystems) for GAd or a CFX connect (Bio-Rad, Hercules, CA, USA) Real Time PCR System for MVA. Transgene-specific primers and probe sequences used for amplifications are listed in Appendix A. Primer and probe sets targeting hHPRT (Hypoxanthine Phosphoribosyltransferase) or h-β-actin were included in each duplex qPCR for detection of host housekeeping DNA and used as an internal control. The expected performance of the multiplex assays was confirmed by comparing quantification cycle (Cq) values of samples that were run both in singleplex and duplex assays on the same 96-well plate. FcDc were determined by absolute quantification using plasmid DNA standard curves for each FSP transgene. Each standard curve, specific for each FSP transgene (FSPA-1 to -4 and FSPB-1 to -4), spanned a range from 5 × 10^1^ to 5 × 10^7^ template plasmid copies. Precision and linearity of the detected signal in the indicated range was assessed for each standard curve by comparing single points Cq, R^2^ and slope values for each curve over different assays. Accuracy was also confirmed by reliable quantification of positive controls with known concentration that were included in each qPCR assay (i.e., Nous-209 vectors clinical lots or defined amounts of FSP template plasmids). Based on standard curves data, the assay limits of quantization were defined as 5 × 10^7^ (high quantitation limit) and 5 × 10^1^ (low quantitation limit) target sequence copies. Importantly, the performance of FSPA-1 to A-4 standard curves was also confirmed during validation of the RT-Q-PCR assay for GAd-209-FSP vectors. Validation of the RT-Q-PCR assay for MVA-209-FSP vectors is ongoing.

### 2.6. Infectivity Assay by Immunostaining

For infectivity assays, test items were serially diluted in DMEM supplemented with 10% FBS, and 100 μL of inoculum was added per well in a 24-well plate containing sub-confluent Hek293 cells for GAd or Vero cells for MVA. At least 3 dilutions were tested. After 48 h, cells were fixed at −20 °C in methanol and then blocked with 1% bovine serum albumin (BSA, Invitrogen, ThermoFisher, Waltham, MA, USA) in PBS. GAd hexon protein was stained by addition of primary mouse anti-hexon polyclonal antibody (1:200 dilution, Abcam, Cambridge, MA, USA) followed by secondary anti-mouse horseradish peroxidase (HRP) conjugated antibody (1:200 dilution, Sigma-Aldrich, Co., St. Louis, MA, USA). MVA was stained by addition of primary rabbit anti-vaccinia virus antibody (1:500 dilution, Abcam, Cambridge, MA, USA) followed by secondary anti-rabbit HRP-conjugated antibody (1:500 dilution, Sigma-Aldrich, Co.). Staining was washed with PBS and revealed with 3,3′ diaminobenzidine (DAB) working solution following the manufacturer’s recommendations for the VECTOR NOVARED Substrate kit. Positive, brown stained cells in the wells from at least two tested dilutions were counted by light microscopy and the results used for calculation of infectious titers as previously reported [22,23]. Notably, the procedures used for infectivity testing of both GAd and MVA vectors follow protocols that have been successfully validated by ReiThera s.r.l. (Rome, Italy).

### 2.7. Mice and Vaccinations

Six-week-old female CB6F1 mice were acquired from Envigo and received day-to-day care from trained personnel at Allevamenti Plaisant SRL. All experimental procedures, including the administration of viral vectors intramuscularly into the quadriceps with a total volume of 100 µL (50 µL per side), were conducted in accordance with approved protocols by the Italian Ministry of Health (Authorizations 213/2016 PR) and adhered to relevant Italian laws (D.L.vo 26/14 and subsequent amendments) under the oversight of the Institutional Animal Care and Ethics Committee of Allevamenti Plaisant SRL.

### 2.8. Ex Vivo Immune Analysis

IFN-γ ELISpot assays were performed on mouse splenocytes, as previously described in [24]. MSIP S4510 plates were first coated with 10 μg/mL of anti-mouse IFN-γ antibody (U-CyTech) and left to incubate overnight at 4 °C. Following washing and blocking steps, freshly isolated mouse splenocytes were then seeded in duplicate at two different cell densities (5 × 10^5^ and 2.5 × 10^5^ cells) and stimulated overnight with peptide pools at a final concentration of 0.4 mg/mL for each peptide. Negative and positive controls were included using the peptide diluent dimethyl sulfoxide (DMSO) and Concanavalin A (ConA), respectively. The following day, the plates were incubated with U-CyTech biotinylated anti-mouse IFN-γ antibody CT655 (dilution: 1/100) followed by conjugated streptavidin-alkaline phosphatase Pharmingen cat n.554065 (dilution: 1/2500) and finally with 5-bromo-4-chloro-3-indoyl-phosphate/nitro blue tetrazolium 1-Step solution (NBT-BCIP solution Thermo Fisher Scientific cat n.34042). An automated plate reader (Immuno Spot CTL, Cleveland, OH, USA) equipped with an enzyme-linked immunosorbent spot (ELISpot) assay video analysis system was employed for plate analysis. Immune responses were assessed based on the quantity of IFN-γ-secreting T cells in reaction to 16 pools of overlapping peptides (P1–P16) representing the polypeptide sequences of the 209 FSP, which are encoded by the vaccine. ELISpot data were reported as IFN-γ Spot Forming Colonies (SFC) per million splenocytes.

### 2.9. Statistical Analyses

Statistical significance was determined by GraphPad Prism (version 10) using the nonparametric, two-tailed Mann-Whitney U test. To confirm correlation between in vivo immunogenicity and RT-Q-PCR potency results for partially and totally inactivated products, the Pearson correlation coefficient was calculated by the PEARSON function in Microsoft Excel between FcDc data and average SFC values across treated mice.

## 3. Results

### 3.1. Nous-209 Vaccine Generation and Immunogenicity

Nous-209 is a multivalent cancer vaccine based on a heterologous prime boost with Great Ape Adenoviral (GAd-209-FSP) followed by Modified Vaccinia Ankara (MVA-209-FSP) vectors. Nous-209 vectors encode 209 frameshift peptides (FSPs), which are tumor-specific neoantigens shared across patients with dMMR cancer [21].

The GAd-209-FSP Drug Product (DP) is composed of four different Drug Substances (DS, four Adenoviral vectors: GAd-209-FSP-A1; GAd-209-FSP-A2; GAd-209-FSP-A3 and GAd-209-FSP-A4) which are identical but for the encoded transgenes (Figure 1a and Appendix A). Each FSP-A transgene has a different nucleotide sequence encoding for an artificial polypeptide, generated by head to tail fusion of a specific subset of the 209 FSPs.

Similarly, MVA-209-FSP DP contains four different DS (four MVA vectors: MVA-209-FSP-B1; MVA-209-FSP-B2; MVA-209-FSP-B3; MVA-209-FSP-B4) each encoding the same FSPs encoded by the corresponding FSP-A transgenes, but arranged in a different order (Figure 1b and Appendix A) [21]. Each FSP-B transgene has thus a different nucleotide sequence, which is also different from the corresponding FSP-A transgenes. Since antibodies are not available for such synthetic transgenes, a human influenza hemagglutinin (HA) tag was added at the carboxyterminal of each polypeptide to explore the option of measuring protein expression (Figure 1).

As part of the quality control testing panel for Nous-209 vaccines, transgene-specific Q-PCRs are performed on each DP lot to confirm the presence of the four DS (Appendix A). The biological function of both components of the vaccine has been previously demonstrated, showing the induction of a T-cell immune response against all the transgenes, both in murine models [21] and in humans enrolled in the clinical trials [13].

### 3.2. Definition of Potency Testing Panel for GAd-209-FSP and MVA-209-FSP Vectors

Despite that the in vivo immunogenicity assay can recapitulate the full vaccine biological activity, such a test was considered not suitable as part of the Nous-209 testing panel for clinical lots release. This is due to the high technical complexity (~1000 peptides are required for testing T cell reactivity against the 209 FSPs) and the high number of animals needed for each testing given the wide inter-animal variability. This complexity is even more relevant considering the proposed prime/boost regimen including the knowledge that MVA vectors are weak for immune response priming, while being optimal boosters [22]. Based on the above limitations, immunogenicity assay was used for product characterization in the initial phases of product development and to support the relevance of the selected in vitro RT-Q-PCR potency assay (see below), but was not included as part of the Nous-209 vaccine potency testing strategy for lots release. In the case of viral vector-based vaccines, the initiation of an immune response in vivo hinges on a series of essential steps. These include effective transcription and transduction of the encoded transgene within transduced cells, subsequent processing of the resultant polypeptide, loading of the relevant peptides onto MHC molecules for presentation and recognition of the resulting surface complex by T-cell receptors. Given this multifaceted mechanism of action, various assays were developed and assessed to determine the most suitable matrix for evaluating the potency of the Nous-209 vaccine. Each of these assays was designed to gauge the vaccine’s biological efficacy at distinct critical stages of the mechanism of action. Transgenes’ specific Q-PCR have been developed and are currently used to confirm the vaccine composition (i.e., the amount of each DS presents in each DP, Appendix A). Importantly, such Q-PCR assays have been confirmed to be specific for the target transgene by absence of detection of the other FSP transgene sequences (Appendix A).

Nous-209 vectors’ ability to efficiently transduce a target cell is also addressed through infectivity testing; the previously described hexon immunostaining assay [23] is currently used to test infectivity of GAd-209 lots, and a similar assay based on anti-vaccinia antibody staining has also been developed for MVA vectors testing. The ratio between physical (vp/mL) and infectious (ifu/mL) titer is also calculated (Appendix A), representing a quality attribute of each vector. Importantly, the infectivity assays being based on staining with anti-vector antibodies, these are not able to discriminate between vectors encoding different transgenes and are thus not suitable for evaluation of the potency of the single DSs when present as a mix in Nous-209 DPs. To allow measurement of transgene expression in infected cells, several attempts were conducted to set-up Western blot (WB) or FACS staining of cells infected with GAd-209-FSP DSs or MVA-209-FSP DSs. Nonetheless, the identification of the synthetic FSP polypeptides proved to be ineffective. For example, WB analysis failed to visualize some of the polyproteins, and in some cases only a diffuse signal appeared both around and below the expected molecular weight. This signal may be attributed to the presence of degradation products. Nevertheless, in vivo immunogenicity for peptides present in each of the four transgenes has been confirmed in mice and humans [13,21], suggesting that all the FSP transgenes encoded by the different DSs are efficiently produced, processed and presented to the immune system by infected cells despite not being efficiently visualized as proteins by standard techniques.

Therefore, to allow a quantitative measurement of transgene expression, an assay based on FSP-transgene-specific Quantitative Reverse Transcription PCR (RT-Q-PCR) on cells infected with Nous-209 vectors was developed.

### 3.3. Setup of a Potency Assay by Quantitative Measure of Transgene mRNA Expression via RT-Q-PCR

To allow quantitative measurement of transgene expression upon infection with GAd-209-FSP or MVA-209-FSP vectors, HeLa cells were selected for assay conduction since they are efficiently infected by Adenoviral and MVA vectors while not allowing significant vector replication.

To specifically detect each of the eight FSP transcripts encoded by Nous-209 vectors, we took advantage of the already developed transgene-specific Q-PCR assays (Appendix A). Oligos and probes targeting endogenous house-keeping transcripts (human β-Actin and human HPRT) were also designed to be used as assay internal controls (Appendix A).

An overview of the potency assay by transgene expression via RT-Q-PCR is depicted in Figure 2 and Appendix A.

Upon infection with either single DSs or with DPs containing the four DSs, transcription of the transgenes occurs over time. RNA is isolated from infected cells, treated with DNAse to eliminate contribute of the viral genome, and reverse-transcribed to cDNA. Four independent duplex Q-PCRs are performed, with oligo/probes/standard curves specific for one of the four FSP transgenes (FSP-A1 to -4 for GAd testing; FSP-B1 to B4 for MVA testing) plus oligo/probes targeting β-Actin for the GAd test or HPRT for the MVA test. The latter represent an internal control in each reaction for both RNA integrity and reverse transcription efficiency, based on the specific Cq (quantification of cycle) value obtained for each sample.

The absolute amount of FcDc detected in a fixed sample volume is calculated for each target transgene in each sample and represents the measure of vector potency. The house-keeping control Cq value is recorded and compared to a reference range. Briefly, an acceptability range is defined for such Cq values based on average ±3X the standard deviation of all values collected over time in different tests (18.7 ± 1 for β-Actin, n = 48; 20 ± 1 for HPRT, n = 16). Samples showing house-keeping Cq values outside the indicated range are excluded from the FcDc data analysis.

### 3.4. Selection of Time Points and MOI

An infection time-course was conducted to define the optimal time point for collection of cells infected with GAd-209-FSP or MVA-209-FSP, with the aim to identify for each vector a time-frame where all the four encoded FSP-A and FSP-B transcripts could be efficiently detected. The time points tested (3 to 16 h for GAd and 1 to 6 h for MVA) were selected based on the specific vector’s biology, with MVA driving transgene expression within minutes after cell entry [25]. Based on data shown in Appendix A, 6 h (GAd) and 3 h (MVA) post infection were selected as collection time points for assay conduction, since the encoded transgenes were expressed with similar kinetics and at very similar levels at these time points. Variations in transcript abundance were noted at prolonged time points likely associated with inherent properties of different factors such as mRNA stability and degradation pathways. These factors are currently the subject of ongoing investigations but are beyond the scope of this study. To address the assay linearity range and select an optimal target MOI for assay conduction, HeLa cells were infected at different MOIs with either GAd-209-FSP (from 20 to 160 vp/cell, Figure 3a) or MVA-209-FSP (from 0.1 to 0.9 ifu/cell, Figure 3b) vectors and collected at the selected time points for transcript levels analysis. The obtained results confirm the assay is linear over the tested MOI ranges for both tested vectors. MOI 80 vp/cell and 0.3 ifu/cell were selected for assay conduction with GAd and MVA vectors, respectively (indicated by arrows in Figure 3).

### 3.5. Evaluation of Assay Variability

To assess the intermediate precision of the method in the established conditions, three independent infections with GAd-209-FSP vectors were performed on three different days and analyzed by two different operators from the same lab. Based on the set-up results, the assay was conducted by infecting cells at 80 MOI with collection at 6 h post infection. Average, SEM and Coefficient of Variation (CV) % were calculated for the obtained results (n = 3, Figure 4a). Similarly, three independent infections were conducted with MVA-209-FSP vectors at 0.3 MOI, with collection at 3 h post infection. Average, SEM and CV% were calculated for the obtained results (n = 3, Figure 4b).

In all cases, CV% did not exceed 21.7%. Overall, the obtained results suggest that the RT-Q-PCR assay is robust and reproducible for both GAd and MVA vectors.

Notably, the full validation of the method was recently conducted for one of the two components, GAd-209-FSP, by ReiThera CDMO. The data collected during validation (Appendix A) are superimposable to those reported here and similar for two GAd-209-FSP clinical lots (Appendix A), further supporting the reliability of the assay at the selected experimental conditions. Importantly, during the validation process the following parameters were successfully addressed: accuracy, precision, specificity, linearity, robustness and range of detection.

### 3.6. The RT-Q-PCR Potency Assay Can Detect Loss of Potency Correlating with In Vivo Immunogenicity Data

The RT-Q-PCR potency assay was assessed for its ability to detect a reduction in potency of the Nous-209 vaccine vectors and, thus, for its suitability for product stability testing. In vitro transcript levels are indeed expected to be reduced in the case of sub-potent vector lots.

Hence, the assay was performed under the following temperature-stressed conditions, which based on preliminary assessment (Appendix A) and published data [26,27] were expected to lead to potency loss for both GAd-209-FSP and MVA-209-FSP DPs. To induce partial potency loss, samples were incubated at +37 °C for 7 days. To completely inactivate both GAd-209-FSP and MVA-209-FSP vectors, samples were incubated at +56 °C for 10 min.

Upon treatment, potency by transgene expression via RT-Q-PCR and potency by infectious virus titer (ifu/mL) were measured and compared to results obtained with the corresponding DPs stored in standard conditions (Figure 5).

The extent of potency loss for each condition and assay was quantified by LRV. The reported results show that both infectivity assay and RT-Q-PCR potency assay are able to detect vector potency loss and to discriminate between complete or partial loss at a similar extent, being thus stability indicators. This is of note considering that the two assays measure two different events required for the vaccine mechanism of action and are conducted in vitro in different cell lines.

To support the relevance of the RT-Q-PCR assay for potency testing, we produced data for one of the two products, GAd-209-FSP, to demonstrate that results obtained in vitro correlate to the immunological potency in vivo. To conduct these experiments, we created a dedicated DP in which only one of the four DS components was partially or completely inactivated. To achieve this, an aliquot of GAd-209-FSP-A3 clinical DS was incubated at two different temperatures to induce partial (51 °C for 3 min, Appendix A) or complete (56 °C for 10 min [26]) loss of potency. An infectivity assay was performed on such samples to confirm the expected impact on vectors’ potency (Table 1). The LRV observed for the two temperature conditions (51 °C, LRV −0.7; 56 °C, LRV −3.4) was similar to that previously described on DP incubated at 37 °C and 56 °C (Figure 5c).

The non-inactivated, partially or totally inactivated A3 vector was used to generate new DPs by mixing single DSs 1:1:1:1:DP: mix of GAd-209-FSP-A1 DS; GAd-209-FSP-A2 DS; GAd-209-FSP-A4 DS; GAd-209-FSP-A3 DS.DP-51 °C: mix of GAd-209-FSP-A1 DS; GAd-209-FSP-A2 DS; GAd-209-FSP-A4 DS; GAd-209-FSP-A3 DS incubated 3′ at 51 °C.DP-56 °C: mix of GAd-209-FSP-A1 DS; GAd-209-FSP-A2 DS; GAd-209-FSP-A4 DS; GAd-209-FSP-A3 DS incubated 10′ at 56 °C.

These three DPs mimic three different possible situations: a totally powerful DP, a DP in which one of the components has partially lost potency, and a DP in which one of the four components is completely inactivated. To ensure that the listed DPs contain the four vector components in the desired ratio (1:1:1:1), the amount of each DS in each of the generated DPs was confirmed by Q-PCR assays specific for each of the FSP-A (one to four) vectors (Appendix A).

Each of the newly generated DPs was tested for potency by transgene expression via RT-Q-PCR and by in vivo immunogenicity testing.

RT-Q-PCR assay readily detected both the partial and the complete potency loss of GAd-209-FSP-A3, even if it was present in the DP mix with fully potent GAd-209-FSP-A1, A2 and A4 components (Figure 6).

FSP-A3 transcript levels expressed by the dedicated DP were indeed similar to those produced by previously tested clinical lots, while they were reduced by about 7.6-fold in DP-51 °C (LRV: −0.9) and almost undetectable in DP-56 °C (LRV: −4.37). Among all samples tested at the different conditions, mRNA expression levels of the other three components of DPs (FSP-A1, FSP-A2; FSP-A4) remained the same (Figure 6).

In vivo immunogenicity testing of the same samples showed induction of similar T-cell response against peptide pools covering FSP-A1, FSP-A2 and FSP-A4 transgenes in mice vaccinated with all the DPs. Differently, T cell response against FSP-A3 was significantly reduced (2.3-fold) in mice receiving DP-51 °C, and abolished in mice vaccinated with DP-56 °C (Figure 7a). In vivo immunogenicity and in vitro RT-Q-PCR assays clearly correlated in their ability to distinguish potency loss of one of the four GAd-209-FSP vaccine components (Figure 7b).

To allow direct comparison of the ability of infectivity, RT-Q-PCR and in vivo immunogenicity testing to detect potency loss, the results obtained by such assays upon testing of GAd-209-FSP-A3 vectors subjected or not to partial or total inactivation are shown in Table 1. Since the infectivity test is not able to discriminate among vectors encoding different transgenes, the infectious titer was assessed on GAd-209-FSP-A3 DS after thermal inactivation and before mixing with other DS to generate the new DPs described above.

The table highlights the ability of the three assays to measure similar percentage of potency loss compared to the fully potent DP.

## 4. Discussion

The quality of a vaccine, including vectored vaccines for both infectious diseases and cancer, must be evaluated by analytical methods addressing its Critical Quality Attributes (CQA). CQA must reflect vaccine identity, purity, structural integrity, and biological activity as a measure of potency, allowing evaluation of lot-to-lot comparability to ensure equivalence of vaccine lots. The particular assays for quantifying these CQA depend on the vaccine platform and the product features [1].

Here we describe the set-up of a potency assay to support the clinical development of Nous-209, a polyvalent cancer vaccine based on heterologous prime/boost administration of a mix of four Adeno (GAd-209) and four MVA (MVA-209) vectors encoding a large number of neoantigens.

The expected biological activity of the vaccine is the induction of T cell immune responses against the large number of neoantigens encoded. Consistently, an immune response to encoded FSP present in each of the four transgenes was detected in vivo after priming with GAd-209-FSP and was boosted by MVA-209-FSP vectors [13,21]. The evaluation of Nous-209 vaccine products’ immunogenicity in vivo requires a very large number of animals because of the complexity linked with its polyvalent nature and the heterologous prime/boost regimen. Hence, the use of in vivo immunogenicity was deemed unsuitable as an analytical method for assessing potency, aligning with the guiding principles for more ethical use of animals in product testing and scientific research (3Rs).

A widely used surrogate potency assay for genetic vaccine is based on in vitro detection of expression of the encoded protein. We included a tag in the synthetic genes to monitor transgene expression based on protein detection (i.e., WB or FACS staining), but the results obtained were inconsistent across the different transgenes and did not correlate with in vivo immunogenicity.

The 209-FSP transgenes encode for artificial proteins generated by head-to-tail fusion of small peptide sequences derived from frameshift mutations in different genes. As such, these artificial proteins do not have any cellular function and are expected to be misfolded rather than have a folding pattern. Misfolded proteins degrade quickly through the proteasome and/or other cellular systems [28], with this potentially resulting in low levels of the full-length protein persisting into the expressing cells. MHC presentation of degradation-derived epitopes is expected to be associated to such processing. In support of the above considerations, inefficient detection of polyepitope proteins (i.e., by Western blot) into expressing cells has indeed been described for rapidly degraded transgene products, despite efficient cell surface presentation of the degradation-derived peptides [29].

Based on the above, we decided to focus on analysis of transgene transcripts levels as the analytical method for assessing potency. Importantly, the polyvalent nature of the vaccine requires an assay able to discriminate the potency of the four Drug Substance (DS) components present in both the DP of GAd-209-FSP(A1-4) and MVA-209-FSP(B1-4).

The transgene specific RT-Q-PCR assay allows simultaneous testing of the single DS present in each DP lot, providing quantitative results for expression of each transgene. The data collected during the set-up demonstrated the assay to be reliable, precise and linear over a wide range of tested MOI for both target vector sets and to be highly specific for each of the vaccine components.

Moreover, here we demonstrated that the assay is stability-indicating, as it is able to detect either total or partial potency loss for both GAd- and MVA-209 vaccine vector components. Notably, RT-Q-PCR and infectivity assay can detect potency loss to a similar extent, thus further supporting the relevance of these two different assays as part of a potency panel. This is not unexpected, since infection and transgene transcription are active processes that require vectors’ full integrity and functionality to efficiently occur [30,31]. However, RT-Q-PCR but not infectivity testing can also discriminate potency loss for single components of the polyvalent vaccine.

Results obtained by RT-Q-PCR testing also correlated to the in vivo detected immunological potency for GAd-209-FSP vectors, further confirming suitability of this test for assessment of vaccine biological activity, in line with regulatory guidelines [2]. Accuracy, sensitivity, specificity and robustness of the RT-Q-PCR assay for GAd-209-FSP testing has been established and documented through validation, further supporting the relevance of the reported correlation.

Based on all the accumulated data, we propose the RT-Q-PCR assay as a relevant in vitro potency assay for Nous-209 vaccine GMP testing, to be part of a potency testing panel that includes also measurement of vectors infectivity, ratio of virus particles/infectious units and vaccine lots composition. To our knowledge similar assay panels have been used for potency testing of approved viral-vectored vaccines [4,5,6,7].

Genetic vaccines encoding neoantigen-based transgenes represent a novel and promising therapeutic option for cancer therapies [19], with a number of examples in clinical development, including off-the-shelf (i.e., Nous-209) and personalized-approach vaccines [15,32]. As for Nous-209, the synthetic nature of the neoantigen transgenes can pose complexity for the development of relevant potency assays based on protein expression.

In conclusion, the RT-Q-PCR assay described here provides a valuable alternative overcoming technical limitations of protein detection and offering a fast and robust procedure, easier to translate to different vaccine constructs including multivalent ones encoding for more than one transgene. Importantly, this assay could be of great value to determine potency also in the context of personalized cancer vaccines for which optimization of protein detection, if even technically possible, may require a timeframe not compatible with the patient’s individualized treatment.

## Figures and Tables

**Figure 1 vaccines-12-00325-f001:**
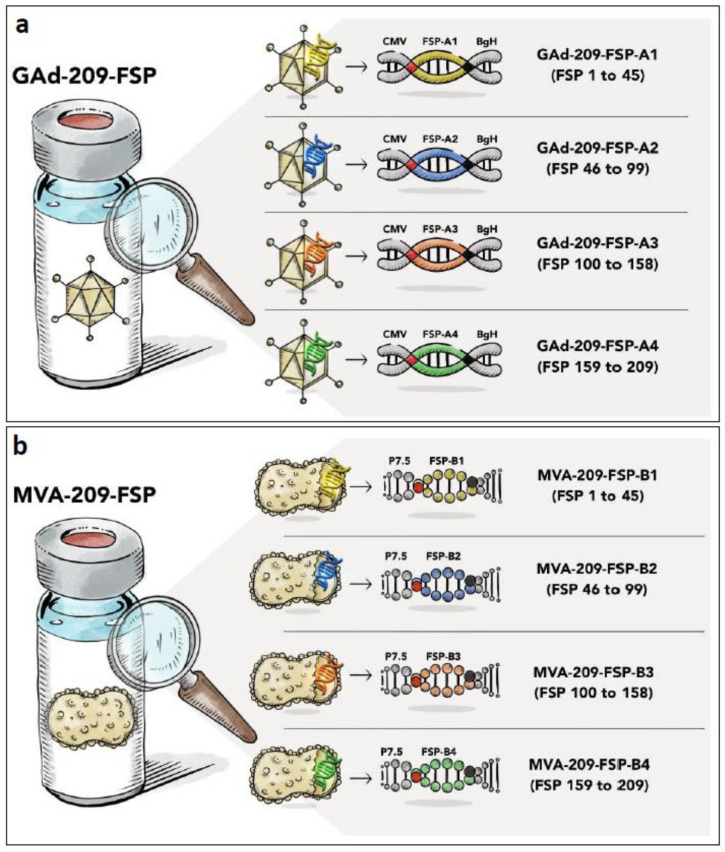
Schematic representation of the composition of GAd-209-FSP and MVA-209-FSP vaccines. (**a**) GAd-209-FSP vaccine is a mix of four GAd vectors, each encoding a synthetic polyepitope consisting of cancer-specific FSP joined head to tail, under the control of human Cytomegalovirus (CMV) promoter (FSP-A1 yellow, FSP-A2 blue, FSP-A3 orange, FSP-A4 green). (**b**) MVA-209-FSP vaccine is a mix of four MVA vectors, each encoding a synthetic polyepitope consisting of cancer-specific FSP joined head to tail, under the control of the early/late P7.5 promoter (FSP-B1 yellow, FSP-B2 blue, FSP-B3 orange, FSP-B4 green). The number of FSP peptides encoded by each transgene is indicated in brackets. Red box/dot: TPA signal peptide. Black box/dot: HA tag.

**Figure 2 vaccines-12-00325-f002:**
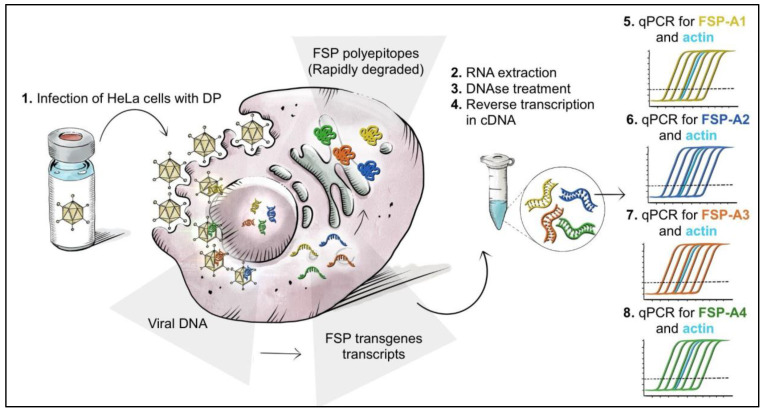
Schematic representation of the RT-Q-PCR transgene expression potency assay for GAd-209-FSP testing. HeLa cells are infected with GAd-209-FSP DP (1), containing four different vectors as depicted. Total RNA is extracted from infected cells (2), treated with DNAse to eliminate vector genome (3) and reverse-transcribed to cDNA (4). Transcript levels for each of the four FSP-A transgenes are assessed by four independent RT-Q-PCRs (5 to 8) with transgene-specific oligo/probe/standard curve sets, indicated by the corresponding colors (FSP-A1 yellow, FSP-A2 blue, FSP-A3 orange, FSP-A4 green). Each Q-PCR assay is a duplex reaction in which, simultaneously to the absolute quantification of the target FSP-A transgene transcript, the β-actin transcript is also amplified with specific oligo/probes (depicted in light blue). The latter is used as housekeeping internal control for RNA integrity and reverse transcription efficiency based on the obtained Cq value.

**Figure 3 vaccines-12-00325-f003:**
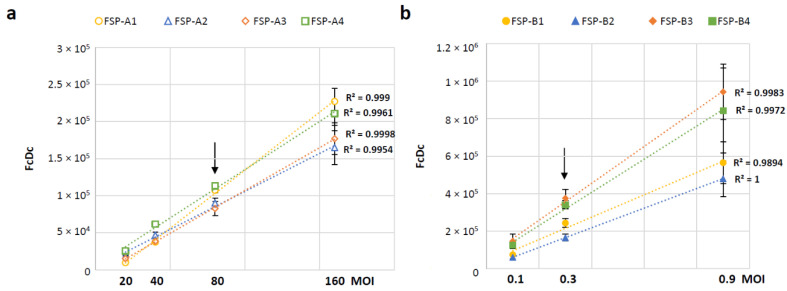
Linearity of FSP transcript levels in HeLa cells infected with GAd-209-FSP or MVA-209-FSP vectors at different MOI. (**a**) HeLa cells were infected at the indicated MOIs with GAd-209-FSP DP. RNA from infected cells was extracted 6 h post infection and reverse-transcribed to cDNA. Transcript levels for each of the four FSP-A transgenes at each MOI were assessed by RT-Q-PCR with transgene-specific oligo/probe/standard curve sets. The number of copies for each FcDc detected in 5 μL of cDNA is shown as average ± standard error mean (SEM) of two independent infections. (**b**) HeLa cells were infected at the indicated MOIs with MVA-209-FSP DP. RNA from infected cells was extracted at the indicated time point post infection and reverse-transcribed to cDNA. Transcript levels for each of the four FSP-B transgenes at each MOI were assessed by RT-Q-PCR with transgene-specific oligo/probe/standard curve sets. The number of copies for each FcDc detected in 5 μL of cDNA is shown as average ± SEM of two independent infections. Linear-fit trendline (dotted) and corresponding R2 are shown for each transcript data set. Arrows indicate the MOI selected for each assay conduction.

**Figure 4 vaccines-12-00325-f004:**
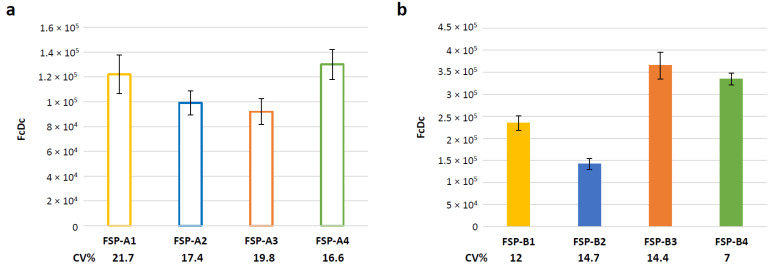
Evaluation of RT-Q-PCR assay variability for potency testing of GAd-209-FSP or MVA-209-FSP vectors. HeLa cells were infected at 80 MOI with GAd-209-FSP (**a**) or 0.3 MOI with MVA-209-FSP (**b**). RNA from infected cells was extracted 6 h (**a**) or 3 h (**b**) post infection and reverse- transcribed to cDNA. Transcript levels for each of the FSP transgenes were assessed by RT-Q-PCR with transgene specific oligo/probe/standard curve sets. The number of copies for each FcDc detected in 5 μL of cDNA is shown as average ± SEM of three independent infections. CV% for each data set is shown below each bar.

**Figure 5 vaccines-12-00325-f005:**
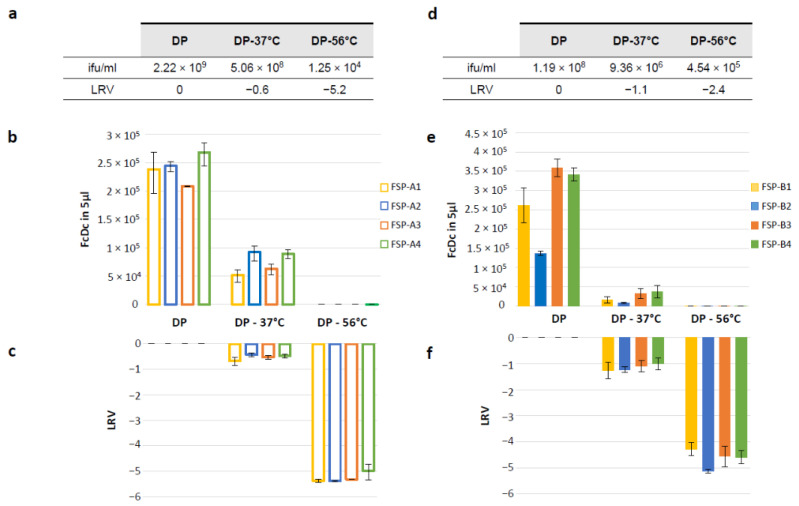
Evaluation of infectious titer and transgenes expression by RT-Q-PCR assay for GAd-209-FSP and MVA-209-FSP vectors after treatment under temperature-stressed conditions. Infectious titer was determined for GAd-209-FSP (**a**) or MVA-209-FSP (**d**) after thawing (DP) or after incubation at +37 °C for 7 days (DP-37 °C) or at +56 °C for 10′ (DP-56 °C). LRV for infectious titer was calculated for each temperature treatment condition compared to DP after thawing. HeLa cells were infected with the same DP samples, and FSP-A (**b**,**c**) or FSP-B (**e**,**f**) transgenes transcript levels were assessed by RT-Q-PCR potency assay. The resulting number of copies for each FcDc is shown (**b**–**e**). LRV was calculated for each transgene and each temperature treatment condition, compared to DP after thawing (**c**,**f**). FcDc and related LRV results are shown as average ± SEM for two independent infections.

**Figure 6 vaccines-12-00325-f006:**
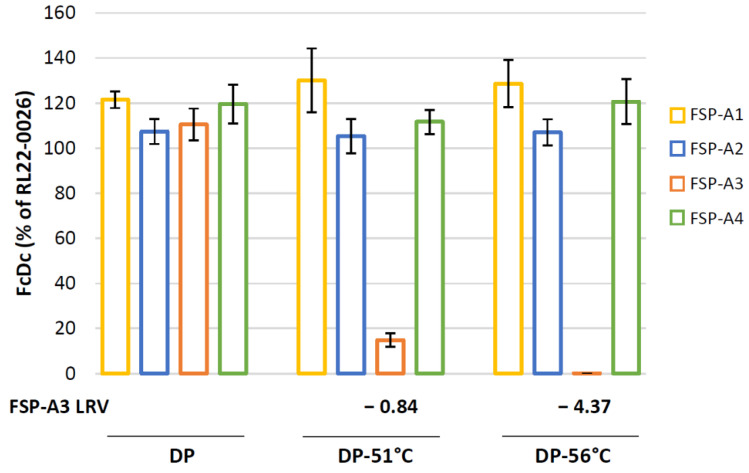
Transgenes expression by RT-Q-PCR for GAd-209-FSP DPs containing inactivated or non-inactivated GAd-209-FSP-A3 vectors. HeLa cells were infected with fully potent GAd-209-FSP DP or with DPs containing partially (DP-51 °C) or fully (DP-56 °C) thermally inactivated GAd-209-FSP-A3 vectors. FSP-A transgenes transcript levels were assessed by RT-Q-PCR potency assay. The resulting number of copies for each FcDc is shown as percentage of the value obtained for the same transcript in cells infected with fully potent GAd-209-FSP clinical DP (lot #RL22-0026) used as assay internal control. LRV for FSP-A3 FcDc results in DP-51 °C and -56 °C was calculated compared to DP. FcDc results are shown as average ± SEM for two (DP-51 °C) and three (DP and DP-56 °C) independent infections.

**Figure 7 vaccines-12-00325-f007:**
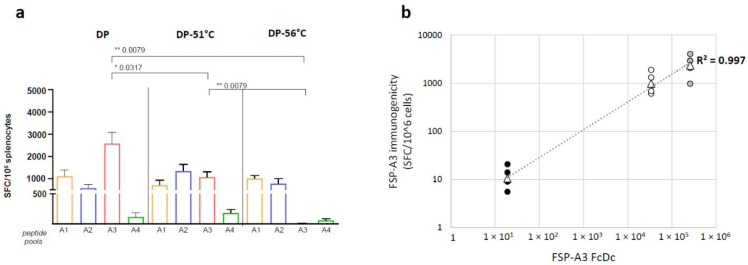
In vivo immunogenicity of GAd-209-FSP with totally or partially inactivated FSP-A3 correlates to RT-Q-PCR potency results. (**a**) CB6F1 mice were immunized with 3 different GAd-209-FSP DPs (total dose of 4 × 10^8^ vp) containing the non-inactivated A3 vector (DP), the partially inactivated A3 vector (DP-51 °C), and the totally inactivated A3 vector (DP-56 °C). Immune responses against each transgene were measured 2 weeks post immunization by ex-vivo ELISpot using peptides pools covering each FSP-A sequence. Data are expressed as number of IFN-γ Spot Forming Cells (SFC)/10^6^ splenocytes. Each bar represents the mean + SEM in 5 mice per group. Statistical comparison was performed by Mann–Whitney test (*p*-values: * *p* < 0.05, ** *p* < 0.01) (**b**) FSP-A3 FcDc as measured by RT-Q-PCR potency assay on the different DPs was plotted against their in vivo FSP-A3 immunological potency (assessed by ex vivo ELISpot using peptides pools covering FSP-A3 sequence). Black circles: individual mice vaccinated with DP-56 °C; white circles: individual mice vaccinated with DP-51 °C; grey circles: individual mice vaccinated with DP. Triangles indicate average SFC for each mice group; linear-fit trendline (dotted) and Pearson correlation coefficient (r) for average SFC vs. FcDc are shown.

**Table 1 vaccines-12-00325-t001:** Comparative analysis of GAd-209-FSP-A3 potency upon thermal inactivation assessed by different assays.

	Infectivity Assay *		RT-Q-PCR **		Immunogenicity In Vivo **	
Sample with	ifu/mL	%	FcDc	%	SFC/10^6^ Cells	%
Fully potent GAd-209-FSP-A3	2.10 × 10^9^	100%	2.41 × 10^5^	100%	2.33 × 10^3^	100%
Partially inactivated GAd-209-FSP-A3	4.14 × 10^8^	19.70%	3.71 × 10^4^	15.40%	9.81 × 10^2^	42%
Fully inactivated GAd-209-FSP-A3	8.40 × 10^5^	0.04%	8.54 × 10^0^	0.003%	1.05 × 10^1^	0.45%

* Infectivity assay was conducted on GAd-209-FSP-A3 DS. ** RT-Q-PCR and immunogenicity testing were performed on DPs generated as described above, and results correspond to those shown in Figure 6 and Figure 7, respectively.

## Data Availability

All data of this study are available in the main text and in Appendix A.

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
