# Peer review of "Development of a Potency Assay for Nous-209, a Multivalent Neoantigens-Based Genetic Cancer Vaccine"

_vaccines, 2024, doi:10.3390/vaccines12030325_

Round 1

Reviewer 1 Report

Comments and Suggestions for Authors

This paper developed and validated an RT-Q-PCR detection method for a multivalent viral vector-based cancer vaccine for vaccine potency evaluation. The study can provide guidance for quality control testing of this new type of vaccine in the industry. However, the paper needs to address the following comments before being recommended for publication.

Comments:

1.      Q-PCR is a classic and widely used in-vitro assay to determine genome copies. What is the major novelty and advantage of this RT-Q-PCR developed?

2.      What are the limitations of this developed potency assay? It claims to detect the viral vector-based vaccine candidates accurately. What if the viral vector is partially filled? Will it give a false positive signal for potency? How is the purity of samples required to get a reliable result?

3.      What is the limit of quantification/detection? Upper and lower limit?

4.      Page 11: The authors claimed both the in vitro infectivity assay and RT-Q-PCR can detect vector potency loss, which correlates to the in-vivo test. Can the author provide a table list for each sample? What are the detected results from the corresponding three assays?

5.      The materials and methods part needs to give more detailed information to guide future readers in setting up their essays:

For examples,

1)Page 2: Line 92: what formulation buffer was used? Will different formulation buffers cause any assay interference?

2)Page 3: Section 2.2: for thermal inactivation, what instrument is used?

3)Section 2.3 cell culture: How the cells were passaged? Either briefly describe or cite papers if it is a published cell culture method.

4)Page 3, 135: Nanodrop instrument type vendor?

5)Page 4: 186-187 automated plate reader instrument details?

6)The LRV calculation equation needs to be given in the methods section. 

Comments on the Quality of English Language

1.   All the Acronyms/Abbreviations need to give the full name when they appear for the first time.

2. All the bulleted points in the manuscript should be changed to a sentence-based paragraph.  

Reviewer 2 Report

Comments and Suggestions for Authors

While the research and its presentation are well presented, it would be pertinent that the author describe their experimental work in light of the specific regulatory guidelines, such as https://www.fda.gov/files/vaccines,%20blood%20&%20biologics/published/Final-Guidance-for-Industry--Potency-Tests-for-Cellular-and-Gene-Therapy-Products.pdf

A major issue with non-biological methods is their validation to ensure their correlation with biological methods. The authors may already have these data to support their findings. I strongly urge adding this to the discussion as well as to the experimental design. 

Comments on the Quality of English Language

Minor changes

Round 2

Reviewer 1 Report

Comments and Suggestions for Authors

The revised version looks good to me and fully addresses my previous comments. I don't have any other comments.